# Palliative Care in Advanced Dementia: Comparison of Strategies in Three Countries

**DOI:** 10.3390/geriatrics6020044

**Published:** 2021-04-22

**Authors:** Shelley A. Sternberg, Shiri Shinan-Altman, Ladislav Volicer, David J. Casarett, Jenny T. van der Steen

**Affiliations:** 1Maccabi Healthcare Services, Modiin 7177806, Israel; sternb_sh@mac.org.il; 2The Louis and Gabi Weisfeld School of Social Work, Faculty of Social Sciences, Bar Ilan University, Ramat Gan 5290002, Israel; shiri.altman@biu.ac.il; 3School of Aging Studies, University of South Florida, Tampa, FL 33620, USA; lvolicer@usf.edu; 4Third Faculty of Medicine, Charles University, 100 00 Prague, Czech Republic; 5Department of Medicine, Duke University, Durham, NC 27710, USA; david.casarett@duke.edu; 6Department of Public Health and Primary Care, Leiden University Medical Center, P.O. Box 9600, 2300 RC Leiden, The Netherlands; 7Department of Primary and Community Care, Radboud university medical center, P.O. Box 9101, 6500 HB Nijmegen, The Netherlands

**Keywords:** dementia, palliative care, cross-cultural comparison, health policy

## Abstract

Palliative care including hospice care is appropriate for advanced dementia, but policy initiatives and implementation have lagged, while treatment may vary. We compare care for people with advanced dementia in the United States (US), The Netherlands, and Israel. We conducted a narrative literature review and expert physician consultation around a case scenario focusing on three domains in the care of people with advanced dementia: (1) place of residence, (2) access to palliative care, and (3) treatment. We found that most people with advanced dementia live in nursing homes in the US and The Netherlands, and in the community in Israel. Access to specialist palliative and hospice care is improving in the US but is limited in The Netherlands and Israel. The two data sources consistently showed that treatment varies considerably between countries with, for example, artificial nutrition and hydration differing by state in the US, strongly discouraged in The Netherlands, and widely used in Israel. We conclude that care in each country has positive elements: hospice availability in the US, the general palliative approach in The Netherlands, and home care in Israel. National Dementia Plans should include policy regarding palliative care, and public and professional awareness must be increased.

## 1. Introduction

Dementia affects the individuals living with it, their families, and the health and social care systems that provide and pay for care [1]. Dementia is associated with a shortened life expectancy [2,3], but often, it is not perceived to be a terminal illness. This limits the adoption of a palliative approach to care [4]. Furthermore, in a Delphi study conducted among experts from 23 countries, participants acknowledged the benefits of palliative care in dementia but did not agree regarding when in the course of the disease it should be implemented [5].

In this paper, we review care that older people with advanced dementia would most likely receive in the United States of America (US), The Netherlands, and Israel with an emphasis on place of residence and access to palliative care, including hospice services, and treatment. These three countries have been highlighted because of differences in the provision of long-term care. For instance, long-term care data of the Organization for Economic Co-operation and Development (OECD) [6] showed that 10% of the population aged 65 years and over in the US received long-term care compared to 13% in The Netherlands and 20% in Israel. In the US, 76% of this long-term care was provided in the community as compared to 66% in The Netherlands and 92% in Israel. In addition, the use of palliative care for advanced dementia also varies between these countries. The US developed hospice models of palliative care for dementia in the 1980s, while the other two countries are just starting to promote models or services in the context of very different cultural attitudes [7,8].

Studying cross-national variation of place of residence and access to palliative care (including hospice care) and treatment of common conditions among older people with advanced dementia can provide useful information for the development of end-of-life care policies and services that meet the needs of people with advanced dementia, their relatives, and society. Furthermore, according to the World Health Organization [9] and Alzheimer Europe [10], there is a need to assess and monitor the use of palliative care as well as to evaluate inequalities in the use of palliative care services in order to design public policies for improvement. In this paper, we present and discuss treatment of a person with advanced dementia and pneumonia in the US, The Netherlands, and Israel.

## 2. Materials and Methods

Articles addressing the care of older people with advanced dementia were reviewed, focusing on the US, The Netherlands, and Israel. Original research and review articles were identified from PubMed. The following search terms were used: (“palliative care” OR “hospice” OR “end of life”) AND (“neurocognitive disorder*” OR “Alzheimer*” OR “dementia” OR “cognitive deterioration” OR “cognitive decline”). We focused on three main issues described and quantified in the literature—(1) data on place of residence, (2) access to palliative (and hospice) care, and (3) treatment of frequent acute conditions. Quality of life in advanced dementia is affected by location of care, which should fit the specific needs of persons with advanced dementia [11,12,13], and provision of palliative care, although the evidence base is smaller than for persons with, for example, cancer [14,15]. Further, important variability in place of residence, various treatments, and outcomes has been highlighted in studies on inequalities in dementia care [16]. Particular and sensitive treatment decisions around hospitalization, artificial nutrition and hydration, and antibiotics commonly need to be made when acute events develop related to advanced dementia, and these treatments might affect comfort or survival [17,18,19].

In this paper, we provide a detailed discussion of a case example of Mrs. S (see Box 1), a woman with advanced dementia and a life expectancy of less than 6 months [20], in order to demonstrate and discuss the different care (place of residence, access to palliative care and treatment of a pneumonia) that she would likely receive in each of the US, The Netherlands and Israel.

Box 1Case scenario.Mrs. S. is an 89-year-old widowed woman with a six-year
history of Alzheimer’s disease. Her past medical history also includes
hypertension, diabetes mellitus, osteoarthritis, and osteoporosis. She has
three children and one of her daughters is her proxy decision maker. She is
now unable to ambulate, can say only a few words, is dependent in all
activities of daily living, does not recognize her family members, and
requires 24 h supervision. She has lost weight in the past 6 months. She is
having trouble swallowing, has been coughing while drinking, and has had one
episode of aspiration pneumonia in the past year. For the last 2 days, she
has been coughing frequently and appears short of breath and sleepy. Upon
physical examination, her physician diagnoses probable recurrent aspiration
pneumonia.

After the authors had inferred from the literature review the care Mrs. S would receive in the three countries, in January–February 2021, we consulted local expert physicians in The Netherlands and Israel with at least three years of experience in treating people with advanced dementia and also affiliated with an academic center in the role of teacher, supervisor or researcher. This consultation was conducted in these two countries only as there are no states considerably varying in policy. In the US, typical care for people with dementia would be strongly state-dependent. In contrast, in The Netherlands and Israel, local experts would be able to oversee typical care for people with dementia in their respective countries. The physicians were recruited through a general call to participate during a teacher and a supervisor meeting of the Leiden academic training center of “elderly care” physicians (caring for older people) in The Netherlands, and through personal invitations to leaders of the other three academic medical training centers. In Israel, a random sample of physicians was recruited through WhatsApp groups of family physicians in the community. They were informed of the purpose of the study and consented to use their evaluation and relevant demographics (gender and experience) for publication while not personally identifiable in the report. We presented the case of Mrs. S and asked them to assess the care Mrs. S would receive in their country. We focused on the same issues and with the same relevant response options as inferred from the literature—choosing place of residence, access to palliative (and hospice) care, and treatment, without providing any results from the literature review before they were asked for their input, to ensure an independent evaluation. We also invited participants to add comments or explanations to their choices. The expert consultation was carried out following the rules of the Declaration of Helsinki of 1975 revised in 2013.

## 3. Results

Mrs. S’s care depended greatly on where she lived. Table 1 describes the care she would receive in the US, The Netherlands, or Israel.

### 3.1. Place of Residence

In the US and The Netherlands, Mrs. S would most likely be living in a nursing home, while in Israel, she would be living in the community [21,22,23,24]. In The Netherlands, long-term care is mostly funded through public funding [25], while in the US and Israel, long-term care is funded through a mix of public and private funding. In Israel, Mrs. S’s multidisciplinary medical homecare team would be provided by one of four publicly administered and financed health funds, whereas her full time live-in personal care worker would be funded through a mix of public and private funds [26].

### 3.2. Access to Palliative (and Hospice) Care

In the US, Mrs. S would be able to receive hospice services in a nursing home that are fully funded by the government. Data from the National Hospice and Palliative Care Organization [27] indicate that patients with a principal diagnosis of dementia had the largest number of days of care on average in 2017, and access to palliative care including hospice care is on the rise. Indeed, in the US, hospice care for nursing home residents serves many residents with dementia [28], and 72% of hospitals were found to have palliative care programs, yet with gaps in access to these programs [29]. Despite this progress, in the US, people living with a serious illness including dementia still do not have reliable access to palliative care, and there is marked variability in program prevalence by hospital type in the US [29].

In contrast, few specific programs provide access to palliative and hospice services for advanced dementia in The Netherlands or in Israel [30,31]. In The Netherlands, a palliative approach to care for nursing home residents is generally used. This is provided by elderly care physicians who are on the staff, yet they rarely (2.5% of deaths) consult palliative care specialists for residents with dementia [32]. In Israel, there is an emerging palliative care discourse in caring for people with advanced dementia living in nursing homes [33]. Furthermore, an Israeli pilot project of home hospice care for advanced dementia was conducted, but this strategy has yet to be widely adopted [31]. It should be noted that, overall, in the three countries, awareness and acceptance among the public that palliative care applies to dementia is still low [34,35,36].

### 3.3. Treatment

End-of-life treatment decisions may include those about artificial nutrition and hydration (ANH), antibiotics, and hospitalization. In the US, decision making is often guided by an individual’s advance directive or proxy preferences. Advance directives may even include refused assistance with eating and drinking in a certain stage of the disease, but the Ethics Committee of the American Medical Directors Association recommended that they should receive comfort feeding despite this advance directive [37]. However, it is possible to honor this advance directive by providing assistance with eating and drinking only when requested by the person with dementia verbally or nonverbally [38].

In the US, the use of ANH varies widely by state, from 2.1 per 1000 older nursing home residents in Utah to 108.3 in Mississippi [38]. Overall, though, the rate of feeding tube insertion in US nursing homes decreased by 50% between 2000 and 2014 [39]. The likelihood of Mrs. S receiving ANH in the US would be highly dependent on her advance directive and proxy decision-makers’ instructions, and the state in which she resided. In addition, Mrs. S would most likely receive antibiotics for her pneumonia [40], with a possibility of hospital transfer again dependent on advance directives and proxy instructions [41,42]. In the US, almost 20% of nursing home residents with advanced dementia were transferred to hospital, mostly for treatment of infections, in the last stages of their lives [42]. In addition, in the US, Mrs. S might even receive mechanical ventilation without substantial improvement in her survival [43].

In The Netherlands, treatments considered futile or burdensome in people with advanced dementia are avoided [44,45]. Therefore, ANH and hospitalization are used rarely, and antibiotic treatment of pneumonia is discouraged [46,47]. The specialist “elderly care physician” on the psychogeriatric ward serves as a gatekeeper to avoid unnecessary interventions [48], while trend analyses show increased treatment to relieve symptoms up to a decade ago [49] and increased antibiotic treatment more recently [50].

A contrasting approach exists in Israel, where Mrs. S would likely receive ANH, be treated with antibiotics, and be hospitalized for pneumonia [51,52]. However, the antibiotic stewardship program in Israel does not recommend antibiotics in patients at the end of life when survival is no longer a goal [53]. This is reflected in the Dying Patient Act of 2005, where certain treatments can be withheld if a patient is deemed to have 6 months or less to live. Fluids can be withheld only in the last 2 weeks of life, and only if they are contributing to the patient’s suffering. In the community, 26% of Israeli people with advanced dementia received ANH [54].

It should be noted that in The Netherlands, about 5% of all older people have an advance directive [55]. Advance care planning usually starts only in the nursing home [56]. In the Northern Americas, including the US and Canada, advance directives are more common than in Europe, but they may not be discussed with family doctors [57,58]. In Israel, although advance care planning is recognized as the means to ensure quality of care, implementation has proved challenging [34].

Overall, the independent expert physicians’ description of typical care for Mrs. S was consistent with the typical care inferred from the literature review. As can be seen in Table 2, physicians were mostly in agreement with the treatment Mrs. S would likely receive in each country. It should be noted that in Israel, physicians believed that a general palliative approach is available for a person with advanced dementia such as Mrs. S. In The Netherlands, while all felt medical care would be provided in the nursing home, where on-staff elderly care physicians are employed, the comments indicated that the expert elderly care physicians perceived that specialist palliative care (as provided by palliative care experts often based in hospitals) would not be needed as they provide generalist palliative care, while others reasoned they provide specialist palliative care themselves. Similar as inferred from the literature from the US (treatment depending on advance directives or proxy decision maker’s choices; Footnote to Table 1), experts from Israel and The Netherlands commented on treatment depending upon family wishes and patient wishes as expressed previously or through advance care planning. The Dutch physicians related this in particular to antibiotic treatment for which most (88%) choose “possibly” (Table 2) and further mentioned medical indicators of prognosis or quality of life as relevant to the treatment decision. Indeed, a study conducted among 69 Dutch nursing home physicians found that prognosis is an important consideration when making antibiotic treatment decisions for patients with dementia and pneumonia [59].

## 4. Discussion

With the aging of the population, end-of-life care for people with advanced dementia is a major healthcare concern. This cross-country comparison indicates that place of residence, access to palliative care including hospice care, and treatment are highly dependent upon particular healthcare systems and cultural context, and therefore, these are relevant to future policy making. The current paper contributes to the literature by highlighting the possible relationship between long-term care systems and the treatment provided in three different countries which also adds to the debate on what is considered good care for people with advanced dementia and to what extent there is in fact a consensus internationally even on care in case of advanced stages of dementia. Many challenges exist given this cross-country variability in care of people with advanced dementia. Firstly, clinical guidelines must take this variability into account [60]. The European Association for Palliative Care (EAPC) has published a White Paper on palliative care for people with dementia [5]. This White Paper advocates the treatment goal of maximization of comfort rather than prolongation of life or maintenance of function, as the person transitions into advanced dementia. Perhaps this focus is not entirely appropriate in all country settings. Clinical guidelines must emphasize the need for palliative care in advanced dementia, while still recognizing cross-country differences.

Secondly, from a policy perspective, half of National Dementia Plans do not include palliative, hospice, or any end-of-life care [10,61,62,63]. These national plans reflect country-specific priorities. The plans of the US and The Netherlands do not emphasize this issue, while it is highlighted in the Israeli plan. To promote a palliative approach to care in advanced dementia, all National Dementia Plans must address this issue in a manner appropriate to their country setting. This may not only improve quality of care but also reduce use of health care resources in realizing most appropriate care for dementia patients [63].

Thirdly, awareness among the general public about the care of older people with advanced dementia, and education and training of health care teams, is relatively limited in all countries. Health care professionals in six countries could not agree on when the stage of dementia appropriate for palliative care actually begins [64], and neither could experts from 23 countries in the EAPC White Paper Delphi panel [5]. While the WHO and other organizations are seeking to raise awareness and equal access to palliative services for all people with chronic life-limiting illnesses, the current review demonstrates that even though availability of palliative care services exists, there is a chance that Mrs. S will not use these services. The reasons for not using palliative care guide future policies regarding this unique care. Strategies to tackle palliative care needs within the population should not only focus on specialist palliative care services but also on guaranteeing adequate palliative care skills for professionals (in nursing homes or home care) and family caregivers [4,65,66,67,68].

A limitation of our work, inherent in a narrative summary, is the lack of quantifiable outcomes or an assessment of the quality of the evidence. The comments by the experts in the consultation were brief, and we could not examine how they perceived access to or availability of a generalist approach to palliative care, in particular how such access would combine with the life-prolonging treatments they choose as typical care in Israel. State-dependent variability in treatment in the US complicates cross-national comparisons. The data may not be fully representative even for small countries with consistent health policy in all areas of the country, such as The Netherlands and Israel. However, the expert consultation was conducted blinded to the results of the literature, and we found the same treatments typical for the countries and presented in publications.

## 5. Conclusions

Care in each country has some positive elements: hospice availability in the US, the use of a palliative approach in The Netherlands, and home care in Israel. Trends in the three different countries show shifts that might continue and trigger a reassessment of advanced dementia care. However, more research on palliative models of care for advanced dementia is needed. Research should also address the earlier stages on which ambiguity may be even greater, and different settings and different countries with different cultural context across the globe. In addition, there is a need for national datasets with standardized data collection. This will facilitate policy decisions, and cross-country comparisons not limited to EU countries of trends over time, treatments, and outcomes for palliative care in advanced dementia.

## Figures and Tables

**Table 1 geriatrics-06-00044-t001:** Care of a person with advanced dementia in the US, The Netherlands, and Israel as inferred from the literature and (in italics) possible revisions from the expert consultation independent from the literature review.

	The US	The Netherlands	Israel
Place of Residence	In a nursing home	In a nursing home (psychogeriatric ward)	At home
Access to Palliative or Hospice Care Services	Hospice	General palliative approach	No/*General* palliative approach ^1^
Treatment			
Artificial nutrition and hydration	State-dependent ^2^	No	Yes ^3^
Antibiotics	Yes ^2^	Possibly	Yes ^4^
Hospitalization	Possibly ^2^	No	Yes
Factors Affecting Treatment Decisions	Advance directives or the proxy decision maker’s instructions	Best interest of the patient judged by a physician with family consult–based on strong tradition of withholding futile care	Dying Patient Act of 2005—legal context rooted in ethical and religious values about life

^1^ Access to a general palliative approach was reported by most Israeli physicians (Table 2). ^2^ Use of these treatments varies mainly depending on advance directives or the proxy decision maker’s instructions. ^3^ Forgoing only in the last 2 weeks of life, and only if they are contributing to the patient’s suffering. ^4^ Forgoing only if survival is no longer a goal of care.

**Table 2 geriatrics-06-00044-t002:** National expert physicians’ evaluations of what kind of care Mrs. S would receive in The Netherlands and in Israel.

	The Netherlands (*n* = 26)	Israel (*n* = 15)
Experts *		
Experience, *n* (%)		
3 to 20 years	12 (46)	10 (67)
20 years and more	13 (50)	4 (27)
Gender, *n* (%)		
women	15 (58)	8 (53)
men	11 (42)	6 (40)
**Typical Care for Mrs. S**		
Place of Residence, *n* (%)		
at home	0	10 (67)
in nursing home	26 (100)	5 (33)
Access to Palliative or Hospice Care Services, *n* (%)		
no	2 (8)	2 (13)
yes, general palliative care approach	18 (69)	10 (67)
yes, hospice or other specialist palliative care services	6 (23)	3 (20)
Treatment, *n* (%)		
Artificial Nutrition and Hydration		
yes	0	8 (53)
yes—region-dependent	0	0
possibly	2 (7)	7 (47)
no	24 (93)	0
Antibiotics		
yes	1 (4)	14 (93)
yes—region-dependent	0	0
possibly	23 (88)	1 (7)
no	2 (8)	0
Hospitalization		
yes	0	8 (53)
yes—region-dependent	0	3 (20)
possibly	0	3 (20)
no	26 (100)	1 (7)

* Missing values for experience: *n* = 1 for The Netherlands and *n* = 1 for Israel; missing values for gender: *n* = 1 for Israel.

## Data Availability

The data that support the findings of this study are available from the authors upon reasonable request.

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
