# Peer review of "Palliative Care in Advanced Dementia: Comparison of Strategies in Three Countries"

_geriatrics, 2021, doi:10.3390/geriatrics6020044_

Round 1
Reviewer 1 Report
The study demonstrates an international comparison of the palliative care and end-of-life treatments for people with dementia in Isreal, the Netherlands, and the United States. The content provides a cross-cultural and inter-health care systems difference and highlights that the clinical guidelines and national plan of palliative care for people with dementia should take this variability into account. The results and discussion is sound for an international readership and clinical implication. However, some points in the current manuscript need further clarification and modification before final acceptance.
- Line 51 What are the reasons you choose the 3 factors- place of residence, access to palliative care (including hospice careà should be after palliative care but not treatments), and treatment, as your target variables in this study? Can you describe the research rationale in more detail with the references that show the comparison for these 3 factors relevant?
- Line 52 ‘treatment’, please define what kind of treatment you refer to here in the paragraph? The term ‘end-of-life treatments’ may be better.
- Line 54, the end of the sentence ‘at large’à not clear
- Line 59-62
It is wired to put a scenario in the Introduction section. You should clarify the research aim in the paragraph by describing that you were going to do the literature review as you mentioned in the Methods section, and also conduct a survey of domestic experts in three countries by using the case scenario. Then the detailed description of the case scenario should be put in the Methods section.
- Line 72-73 ‘independent and blinded to the care the authors from the three countries 72 had inferred from the literature’à Not clear, need English editing.
- Line 76-79,
Is the reason that you only survey physicians in the Netherland and Israel ‘In the US, typical care for people with dementia would be strongly state-dependent. In contrast, in the Netherlands and Israel, local experts would be able to oversee typical care for people with dementia in their respective countries.’? Please clarify in the content
- ‘Ms. S’à ‘Mrs. S’, please make sure it consistent in whole the manuscript.
Author Response
Please see attachment for full response letter
Reviewer 1.
The study demonstrates an international comparison of the palliative care and end-of-life treatments for people with dementia in Israel, the Netherlands, and the United States. The content provides a cross-cultural and inter-health care systems difference and highlights that the clinical guidelines and national plan of palliative care for people with dementia should take this variability into account. The results and discussion is sound for an international readership and clinical implication. However, some points in the current manuscript need further clarification and modification before final acceptance.
Response: thank you for your appreciating relevance of the work for an international audience.
- Line 51 What are the reasons you choose the 3 factors- place of residence, access to palliative care (including hospice care should be after palliative care but not treatments), and treatment, as your target variables in this study? Can you describe the research rationale in more detail with the references that show the comparison for these 3 factors relevant?
Response: Thank you for raising this important conceptual and methodological point. Treatment indeed refers to specific treatments. In this case scenario, it refers to treatment of the pneumonia that is related to the condition of advanced dementia. Quality of life in advanced dementia is affected by location of care which should fit the specific needs of persons with advanced dementia, and by provision of palliative care. The three modalities of treatments of the pneumonia (artificial nutrition and hydration, antibiotics and hospitalization) often involve difficult decisions that are common in advanced dementia and the treatments may affect comfort or might also affect survival.
We inserted references to the literature to support choice of the 3 factors in the Methods section, and we corrected and clarified “treatment” in line 51-53 of the Introduction.
- Line 52 ‘treatment’, please define what kind of treatment you refer to here in the paragraph? The term ‘end-of-life treatments’ may be better.
Response: thank you.
We clarified by adding “of common conditions” to treatment.
- Line 54, the end of the sentence ‘at large’ not clear
Response: we did not identify a problem with the sentence that ended line 54 (“… for the development of end‐of‐life care policies and services that meet the needs of people with advanced dementia, their relatives, and society at large.”). However “at large” can be missed, and perhaps “public health functions” in the next sentence was not clear (“Furthermore, according to the main public health functions outlined by the World Health Organization…”)
We omitted “at large” and “the main public health functions outlined by”
- Line 59-62. It is wired to put a scenario in the Introduction section. You should clarify the research aim in the paragraph by describing that you were going to do the literature review as you mentioned in the Methods section, and also conduct a survey of domestic experts in three countries by using the case scenario. Then the detailed description of the case scenario should be put in the Methods section.
Response: Thank you.
We explained the aim of the paper in more general terms in the last paragraph of the Introduction, and we moved the case scenario to the Methods.
- Line 72-73 ‘independent and blinded to the care the authors from the three countries 72 had inferred from the literature’à Not clear, need English editing.
Response: it would be important to clarify this.
We rephrased saying that we conducted the literature review first, and we consulted the experts afterwards. Toward the end of the paragraph, we added that we did not provide any findings from the literature to the experts before asking them for their input, to ensure their evaluation was independent from the authors’ review.
- Line 76-79, Is the reason that you only survey physicians in the Netherland and Israel ‘In the US, typical care for people with dementia would be strongly state-dependent. In contrast, in the Netherlands and Israel, local experts would be able to oversee typical care for people with dementia in their respective countries.’? Please clarify in the content
Response: indeed, this was our line of thought.
We clarified that we conducted the expert consultation in two countries only.
- ‘Ms. S’à ‘Mrs. S’, please make sure it consistent in whole the manuscript.
Response: Thanks for catching this inconsistency.
We replaced Ms., using Mrs. consistently.

Reviewer 2 Report
An international team in the field of palliative care in Israel, US, and the Netherlands perform a cross-cultural comparative analysis of health policies and level/forms of its implementation in advance dementia focusing on three a priori major domains: where (setting), when(access), how(treatment). The analysis unveils the positive aspects of each scenario and warns about insufficient consensus on what is probably the most critical of the three aspects: the treatment (nutrition and hydration). They use a case example to provide a clear insight of how the same person would receive care depending on the country, according to expert consultations. The work provides arguments to allow the authors to demand national dementia plans to include end-of-life palliative care, which is actually limited because dementia is not considered a terminal disease despite shorter life expectancy. They also point at the need to increase general and professional awareness of this topic.
Some questions need to be addressed
1) Line 66. "using search terms for palliative care and hospice, and dementia 67 and Alzheimer disease. "
The specific keywords used in the literature search must be provided.
2) Line 68. The three main issues are important, but the rationale for their choice and not the inclusion of others should be provided.
3) Line 77. Despite "In the US, typical care for people with de-77 mentia would be strongly state-dependent." this does not imply that local expert physicians in the US have an opinion in this respect. How this (not assessing the care Mrs S would receive in US) may have affected the output of the analysis?
4) Line 136. " However, what is 136 possible is assistance with eating and drinking provided only when requested by the per-137 son with dementia" Please, provide some clarification with this respect (the 'only when requested') since the work refers to an end-of-life stage and aphasia can be present. Does it refer to the ' living will or last wills' declarations?
5) Line 122, 152 and others . "elderly". The term elderly is considered ageist. Please, change to older person or equivalent.
6) Table 2. Gender shouldn't be binary. Too late now. Please, consider this next time.
7) Line 196. Cultural context is stated as a factor. To which extent religious/spiritual aspects or the familyist tradition of each country (Israel) underly these differences. How the 'clinical guidelines' interact with the religious/spiritual/believes and wills of the person with dementia and their families?
8) Lines 217-229. This paragraph is of paramount relevance. The challenge but also the "reasons for not using palliative care guiding future policies regarding this unique care" have an important impact.
"…..the current review demonstrates that even 223 though the availability of palliative care services exists, there is a chance that Mrs. S will not 224 use these services. The reasons for not using palliative care guide future policies regarding 225 this unique care. For example, strategies to tackle palliative care needs within the popula-226 tion should not only focus on specialist palliative care services but also on guaranteeing 227 adequate palliative care skills for professionals (in nursing home or home care) and family 228 caregivers4,55."
The ' For example,' contributes to underestimating the second part of the sentence, which could be provided as a direct statement. Reference 4 refers to physicians' reasons, and Reference 55 refers to nurses' reasons. The authors add the need also to consider the family actors (family caregivers)
I'd suggest to omit 'for example':
The reasons for not using palliative care guide future policies regarding 225 this unique care. Strategies to tackle palliative care needs within the popula-226 tion should not only focus on specialist palliative care services but also on guaranteeing 227 adequate palliative care skills for professionals (in nursing home or home care) and family 228 caregivers4,55."
9) Lines 230-238. The limitation statement highlights the shortcomings of the approach: the narrative. I'd suggest not increasing the limitations by using the 'A tale' in the title since it worsens the confidence. I'd suggest using the scientific approach, such as the authors refer to in line 230. Something like ' Palliative care in advanced dementia: A cross-cultural narrative summary from experts." or similar.
10) Lines 243-246 and 246-249. Are long sentences that could be structured as a kind of decalogue. Please, make a statement of the needs foreseen in 1) 2) 3) or factors to be considered.
Minor. Table 1.
In a nursing home (psy-
chogeriatric ward)
would look better as
In a nursing home
(psychogeriatric ward)
Similarly
the proxy decision mak-
er's instructions
the proxy decision
marker's instructions
Check many others like this
Also the gap . alienation in the Israel columns
Author Response
Please see attached file for full response letter.
Reviewer 2.
An international team in the field of palliative care in Israel, US, and the Netherlands perform a cross-cultural comparative analysis of health policies and level/forms of its implementation in advance dementia focusing on three a priori major domains: where (setting), when(access), how(treatment ). The analysis unveils the positive aspects of each scenario and warns about insufficient consensus on what is probably the most critical of the three aspects: the treatment (nutrition and hydration). They use a case example to provide a clear insight of how the same person would receive care depending on the country, according to expert consultations. The work provides arguments to allow the authors to demand national dementia plans to include end-of-life palliative care, which is actually limited because dementia is not considered a terminal disease despite shorter life expectancy. They also point at the need to increase general and professional awareness of this topic.
Some questions need to be addressed
Response: thank you for recognizing the three domains we addressed and implications of our work for dementia policies.
1) Line 66. "using search terms for palliative care and hospice, and dementia 67 and Alzheimer disease. "
The specific keywords used in the literature search must be provided.
Response: we added the specific search terms we used: (“palliative care” OR “hospice” OR “end of life”) AND (“neurocognitive disorder*” OR “Alzheimer*” OR “dementia” OR “cognitive deterioration” OR “cognitive decline”).
2) Line 68. The three main issues are important, but the rationale for their choice and not the inclusion of others should be provided.
Response: Thank you for raising this important point, which was raised by the other reviewer as well. Treatment indeed refers to specific treatments. In the case scenario, it refers to treatment of the pneumonia that is related to the condition of advanced dementia. Quality of life in advanced dementia is affected by location of care which should fit the specific needs of persons with advanced dementia, and provision of palliative care. The three modalities of treatments of the pneumonia (artificial nutrition and hydration, antibiotics and hospitalization) often involve difficult decisions that are common in advanced dementia and the treatments may affect comfort or might also affect survival.
We inserted references to the literature to support choice of the 3 factors in the Methods section.
3) Line 77. Despite "In the US, typical care for people with de-77mentia would be strongly state-dependent." this does not imply that local expert physicians in the US have an opinion in this respect. How this (not assessing the care Mrs S would receive in US) may have affected the output of the analysis?
Response: we acknowledge that state-dependent variability in care complicates cross-national comparisons. On the other hand, the US offers the largest body of literature on treatment of persons with advanced dementia.
We added the complication of cross-national comparisons to the paragraph on limitations in the Discussion.
4) Line 136. " However, what is 136 possible is assistance with eating and drinking provided only when requested by the per-137 son with dementia" Please, provide some clarification with this respect (the 'only when requested') since the work refers to an end-of-life stage and aphasia can be present. Does it refer to the ' living will or last wills' declarations?
Response: This strategy for dealing with persons who are dependent for eating and drinking refers only to those persons who stated in their advance directives that they do not want to survive to the stage of advance dementia and do not want to be helped with eating and drinking.
We recognize that the request for help could be non-verbal and clarified this point in the text.
5) Line 122, 152 and others . "elderly". The term elderly is considered ageist. Please, change to older person or equivalent.
Response: we apologize and we regret the choice of the Dutch professional organization to replace “nursing home physician” by “elderly care physician” as the official term for their profession. We do not use the term elderly except for in this case when the organization does not leave choice.
In the Methods where we mention elderly care physicians for the first time, we inserted quotation marks and between brackets, we explain these physicians care for older people.
6) Table 2. Gender shouldn't be binary. Too late now. Please, consider this next time.
Response: thank you, we are indeed using more than two response options in most new research.
7) Line 196. Cultural context is stated as a factor. To which extent religious/spiritual aspects or the familyist tradition of each country (Israel) underly these differences. How the 'clinical guidelines' interact with the religious/spiritual/believes and wills of the person with dementia and their families?
Response: we agree that religion may be important in shaping ethical values.
We added religion to Table 1, column Israel, factors affecting treatment decisions.
8) Lines 217-229 . This paragraph is of paramount relevance. The challenge but also the "reasons for not using palliative care guiding future policies regarding this unique care" have an important impact.
"…..the current review demonstrates that even 223 though the availability of palliative care services exists, there is a chance that Mrs. S will not 224 use these services. The reasons for not using palliative care guide future policies regarding 225 this unique care. For example, strategies to tackle palliative care needs within the popula-226 tion should not only focus on specialist palliative care services but also on guaranteeing 227 adequate palliative care skills for professionals (in nursing home or home care) and family 228 caregivers4,55."
The ' For example,' contributes to underestimating the second part of the sentence, which could be provided as a direct statement. Reference 4 refers to physicians' reasons, and Reference 55 refers to nurses' reasons. The authors add the need also to consider the family actors (family caregivers )
I'd suggest to omit 'for example':
The reasons for not using palliative care guide future policies regarding 225 this unique care . Strategies to tackle palliative care needs within the popula-226 tion should not only focus on specialist palliative care services but also on guaranteeing 227 adequate palliative care skills for professionals (in nursing home or home care) and family 228 caregivers4,55."
Response: thank you for your helpful comment.
We deleted “for example” as suggested. We added references that indicate the importance of educating families as well.
9) Lines 230-238. The limitation statement highlights the shortcomings of the approach: the narrative. I'd suggest not increasing the limitations by using the 'A tale' in the title since it worsens the confidence. I'd suggest using the scientific approach, such as the authors refer to in line 230. Something like ' Palliative care in advanced dementia: A cross-cultural narrative summary from experts ." or similar.
Response: Thank you for your comment.
We modified the title of this paper to “Palliative care in advanced dementia: Comparison of strategies in three countries”
10) Lines 243-246 and 246-249. Are long sentences that could be structured as a kind of decalogue. Please, make a statement of the needs foreseen in 1) 2) 3) or factors to be considered.
Response: Thank you.
We split the last two run-on sentences in the Conclusion in two.
Minor.
(11) Table 1.
In a nursing home (psy-
chogeriatric ward)
would look better as
In a nursing home
(psychogeriatric ward)
Response: the problem occurred with typesetting of the manuscript.
We could easily fix by moving the word to the next sentence.
(12) Similarly
the proxy decision mak-er's instructions
the proxy decision
marker's instructions
Response: Thank you.
We moved to the next sentence.
(13) Check many others like this
Response: than you.
We checked and revised for clarity if needed.
(14) Also the gap . alienation in the Israel columns
Response: Thank you.
We fixed the gap in the Israel Columns.
